

# Expansion of sandhill cranes (*Grus canadensis*) in east Asia during the non-breeding period

Linqiang Gao[1], Chunrong Mi[2,3] and Yumin Guo[1]

[1] College of Nature Conservation, Beijing Forestry University, Beijing, China
[2] Key Laboratory of Animal Ecology and Conservation Biology, Institute of Zoology, Chinese Academy of Sciences, Beijing, China
[3] University of Chinese Academy of Sciences, Beijing, China

Corresponding author
Yumin Guo, guoyumin@bjfu.edu.cn

## ABSTRACT

**Aim**. Historically, the distribution of Sandhill Cranes included much of North America and extending in summer into northeast Russia. In recent years, observations of sandhill cranes in Asia during the non-breeding period have been frequently reported. However, the distribution and abundance of sandhill cranes during the non-breeding period in Asia have rarely been summarized and studied. Our study aimed to analyze the status of sandhill cranes that have spread south into East Asia during the non-breeding period and to assess the possible impacts of their potential spread.

**Methods**. Based on opportunistic data collected in the field and occurrence data collected online over the past half century, we used Geographic Information System to visualize the spatial distribution changes and regression analysis to analyze and visualize the changes in the amount of individuals over time.

**Results**. In the last 50 years, the distribution of sandhill cranes during the non-breeding season in Asia spanned 31 degrees in longitude to the west and 15 degrees in latitude to the south. Their distribution in Asia has expanded to 17 provinces and municipalities in China, Japan and South Korea. The amount of cranes in the non-breeding period in Asia increased significantly from 1963 to 2017. According to the historical records in East Asia, sandhill cranes were mixed with five other species of crane groups.

**Main conclusions**. These results indicate that the range and amount of sandhill cranes have expanded. Sandhill cranes were mixed with five other crane species, which indicate their adaptability to a range of habitat types and food resources. The implications of these trends in sandhill cranes in East Asia for this and other crane species warrants further research.

## INTRODUCTION

The sandhill crane (*G. canadensis*) current perhaps the most abundant of the 15 species of cranes, with a global population of 670,000 to 830,000 in 2018, and it is still increasing (*Wetlands International, 2018*). The International Union for Conservation of Nature (IUCN) listed the sandhill crane as a species of Least Concern (*IUCN, 2017*). Sandhill cranes are widely distributed in North America (*Johnsgard, 1983*; *Johnsgard, 2008*). Historically,

the wintering and breeding areas of sandhill cranes were limited to North America, with some flying over the Bering Strait to Asia and breeding in northeastern Russia (*Walkinshaw, 1950*; *Johnsgard, 1983*). The distribution map of sandhill cranes in North America is shown in Fig. A1.

Among the large populations of sandhill cranes, more than 99% of individuals have migratory habits, and the remaining population is not migratory (*Wetlands International, 2018*; *USFWS, 2018*). According to genetic characteristics, the migrating population can be divided into two subspecies: lesser sandhill cranes (*G. canadensis canadensis*) and greater sandhill cranes (*G. canadensis tabida*) (*Rhymer et al., 2001*; *Wetlands International, 2018*). The sandhill cranes of the different subspecies in the Mid-continent population (MCP) have different morphologies. For management purposes, migratory sandhill cranes have been grouped into six populations by the U.S. Fish & Wildlife Service (USFWS), despite their subspecies affiliation (*Hanna, Schummer & Petrie, 2014*). The MCP is comprised of both subspecies of sandhill cranes, whereas the other populations are comprised almost exclusively of either the greater or lesser subspecies (*USFWS, 2018*). It is the lesser sandhill crane subspecies that migrates to East Asia from central North America (*Johnsgard, 1983*; *Jones et al., 2005*; *Krapu et al., 2011*; *Krapu et al., 2015*). These individuals are part of the largest population of sandhill cranes (MCP), and that population continues to increase in amount and its breeding range in Asia (*Dubovsky, 2016*; *Dubovsky, 2017*).

In the past, the tracking research on sandhill cranes in North America has partially revealed its breeding grounds in Asia. The tracking results showed that approximately 23% of the 133 individuals tracked in the MCP fly to Russia during breeding season, most of which were distributed in Chukotka, and a small amount was distributed in Yakut and Koryakia. The spring migration period of the sandhill cranes flying to Russia during the breeding season and wintering in North America lasted an average of 73 days. The cranes arrived at the breeding grounds in mid-May and stayed at the breeding grounds for an average of more than 142 days and left in early September. The autumn migration was much faster than the spring migration, and the sandhill cranes left Russia within 7 days and then moved over 39 days to North America, eventually reaching the wintering grounds in mid-October and staying an average of 46 days (personal communication with Bysykatova Inga Prokopievna, 2017). Most of the breeding cranes were confined to within 75 km of the coast (*Krapu & Brandt, 2005*). The migration rhythm of the sandhill cranes that breed in Russia and winter in China, Japan and Korea has not yet been studied.

It is speculated that in the 1980s, the total amount of sandhill cranes in the Soviet Union (Russia) may have reached 20,000 (*Johnsgard, 1983*). Since the early 1990s, sandhill cranes have expanded their distribution to Yakutia with an increased amount of individuals (personal communication with Bysykatova Inga Prokopievna, 2012). In Yakutia, the amount of sandhill cranes increased by 1.8 times from 1984 to 1994 (*Degtyarev, 2008*); from 1988 to 2008, the breeding area of the sandhill cranes moved westward by 200–250 km; from 1995 to 2009, the amount of breeding sandhill crane individuals in the tundra region near the Indigirka River increased by 13.3 times (*Prokopievna, 2012*).

Sandhill cranes wintering in East Asia have been reported since 1963, with first reports in Izumi, Japan (*Crane Park Izumi, 2017*). In 1970, a sandhill crane was recorded mixing
with red-crowned cranes (*G. japonensis*) in Hokkaido, Japan. In 1979, sandhill cranes were recorded for the first time in Jiangsu Province, China (*Masatomi, 1972*; *Kuang, Xian & Wang, 1981*). In recent years, an increasing amount of sandhill cranes have been recorded in China, Japan and Korea during non-breeding periods (*Harris & Mirande, 2013*; *Cheng et al., 2014*; *Crane Park Izumi, 2017*). In the winter of 2017, only 7 sandhill cranes were recorded in Japan (*Crane Park Izumi, 2017*). It has been suggested that the status of the sandhill cranes in China should be defined as rare winter birds (*Cheng et al., 2014*). The IUCN Red List has also changed its status in China, Japan, and South Korea from straggler birds to a naturally distributed species (*IUCN, 2017*).

However, the temporal and spatial characteristics of the distribution of sandhill cranes in East Asia during the non-breeding period have not yet been summarized and analyzed systematically. With that aim, we collected data from the field, online and the spatial and quantity changes of the cranes that have been analyzed over time. Our study clarifies the changes in the distribution and the amount of sandhill cranes in East Asia during the non-breeding period in the past half century.

## MATERIALS & METHODS

### Data collection:

The data sources for this study consist of five parts.

### *Field data source*

During the 8 years from 2008 to 2015, we collected the amount and the distribution data of sandhill cranes while conducting field surveys of other crane species in China and Japan.

From 2008 to the present (2018), we carried out field investigations of Northeast Asian cranes, including hooded cranes (*G. monacha*), white-naped cranes (*G. vipio*), red-crowned cranes, Siberian cranes (*G. leucogeranus*), demoiselle cranes (*Anthropoides virgo*) and Eurasian cranes (*G. grus*) in the spring, autumn and winter. The spring survey time was from mid-March to the end of April, and the autumn survey time was from mid-September to mid-November. The locations were on the edge of the Songnen Plain and the Xiaoxing'an Mountains in Northeast China (Zhalong, Xinqing, Dazhan River, and Lin Dian, Momo Ge, Xiang Hai, etc.). The winter survey period was from the end of November to the following March in the middle and lower reaches of the Yangtze River (Jiangsu, Anhui, etc.). In the winters of 2010 and 2014, we visited Japan for a short period of time to understand the wintering situation of the hooded crane. Using the telescope direct observation method for all the field research on the cranes in Northeast Asia, we have been paying attention to the presence of sandhill cranes and recording them in time.

### *Paper source*

By searching the "Chinese Knowledge Network" (http://www.cnki.net); "Web of Science" (http://apps.webofknowledge.com/) and other database systems, we collected new records of sandhill cranes released in and before 2017 by searching "sandhill cranes". Additionally, accessible data on the location, time, and quantity of sandhill cranes were collated. All of the new records were included in our research.

### News and bird watching websites source

Through searching domestic and international websites, we identified news and bird-watching records of sandhill cranes. Using the Chinese search engine "Baidu" (https://www.baidu.com/) and other online public platforms, sighting records and news of sandhill cranes were collected. The sighting records and news of sandhill cranes in Korea were gathered through online public platforms such as the Korean search engine "NAVER" (https://www.naver.com/). For the records in Japan, the Google search engine (https://www.google.com/) was used. All the searches were based on the words "sandhill cranes" and "news". Some of the bird watching information provided by birdwatchers was verified with photos. For the records without photos, we first judged whether there was more than one distribution record of sandhill cranes in this area, and then filtered data by determing whether the site was suitable for cranes. If it was true, the record was used in this study. All the information for the data analysis in this paper was used after deduplication.

### Global biodiversity information facility source

The Global Biodiversity Information Facility (GBIF) (http://www.gbif.org/) is an open data research site funded by various governments that provides data on species across the world. We downloaded the distribution data of sandhill cranes in Asia and excluded Russia (which is a breeding area for the species) from 1950 to 2017.

### Crane museum of Izumi, Japan

We collected the time and amount information of sandhill cranes recorded in Izumi, Kagoshima City, Japan from 1963–2017 on the website of "Japanese Izumi Crane Museum" (http://www.city.kagoshima-izumi.lg.jp/).

## Data analysis
### Data filter

We deleted data with incomplete information (missing the year or the place of sighting) and obtained the latitude and longitude in Google Maps using the name of the recorded place. Duplicated data were removed by checking the time, place, and recorder information. The final data included time (year/month/day), year (winter), the name of the place, the latitude and longitude, the sources of information, the age of the individuals, whether a family group was recorded, whether the sandhill crane was mixed with other cranes and what crane species it was mixed with, and other additional information. If the data included information on month, and the month of observation was in January-May, then the record was assigned the previous year, to indicate the bird(s) was considered to have been present in East Asia that entire winter. The data without month information were recorded as the winter of the current year.

We also recorded the geographical location and the amount of species that the cranes were mixed with; the year and the geographical location that only included subadult individuals; and the year and geographical location in which the family groups appeared.

### Distribution change analysis

The data were classified according to the year (due to the large time span, the amount of data was relatively small before 2000 and the data were not evenly divided by year).

The data were artificially attributed to three different periods: before 2000 (1963–1999), 2000–2009, and 2010–2017. We used ArcGIS 10.1 to created maps for the different periods of time.

***Quantity changes in successive years***

When data were recorded multiple times in the same place within the same year, only one piece of data with the largest amount of individuals was kept while the others were deleted. The years from 1963 to 2017 with no records were included with the amount of observations was defined as zero. At the same time, the data with observation records but with no exact quantitative data were defined as 1. Finally, we created a counting table of the sandhill cranes during the non-breeding period in Asia (see data source in Table A1).

Different power polynomial regressions were performed on the year (x) and the amount of individuals (y). To obtain the overall fluctuation trend of the sandhill cranes over the past 55 years and to reduce the impact of data differences in individual years on the overall trend fitting, a simple mean value filter was applied to the original data to reduce the impact of the overall trend from a single piece of data. We took the filter window length as $n = 10$ years (taking the mean data value from 1963-1972 as the amount of cranes in 1967) and the specific step length as $n = 1$ year (the step length in 1 year). We used Python 3.6.1 to conduct a mean value filter and calculate the coefficient of determination, the "numpy" packages for the polynomial fitting, and the "matplotlib" packages for the data visualization.

## RESULTS

During the 55 years from 1963 to 2017, a total of 169 valid data points (including locations and years simultaneously) were obtained for the distribution of sandhill cranes.

The results of the field survey were as follows. In 2009, one sandhill crane was found in Xinqing Wetland Park in Yichun, Heilongjiang Province. In 2011, 2014 and 2015, one sandhill crane was found in the Zhalong Nature Reserve in Heilongjiang Province each year. In 2015, one sandhill crane was observed in Caizi Lake, Anhui Province. In Izumi, we observed sandhill cranes mixed with hooded cranes and had photos. The results of this part of the field work survey are also included in the website of the Crane Museum of Izumi. In February 2010 and January 2014, 1 to 3 sandhill cranes were observed in Izumi, Japan. During the 6 years of 2008, 2009, and 2011 to 2014, 1 ~ 4 sandhill cranes were observed and photographed in the Yancheng National Wetland Nature Reserve for Rare Birds in Jiangsu province. The other four methods obtained 21, 44, 52, and 38 pieces of data, respectively, as shown in Table 1.

### Mixed, subadult, adult and family records

There are 29 pieces of data clearly record whether the sandhill cranes were active independently or mixed with other crane species among all the data. The data shows that sandhill cranes were only observed with other cranes species, and never reported away from other crane species during the non-breeding period in east Asia. They were reported to have mixed with the following five species of cranes distributed in the same period:

**Table 1 Data source table.** Different periods of data corresponding to different sources.

| Period | Field | Literature | News | GBIF | Izumi | Total |
|---|---|---|---|---|---|---|
| Pre 21C | 0 | 6 | 1 | 3 | 20 | 30 |
| 2000~09 | 4 | 6 | 16 | 17 | 10 | 53 |
| 2010~17 | 10 | 9 | 27 | 32 | 8 | 86 |
| Total | 14 | 21 | 44 | 52 | 38 | 169 |

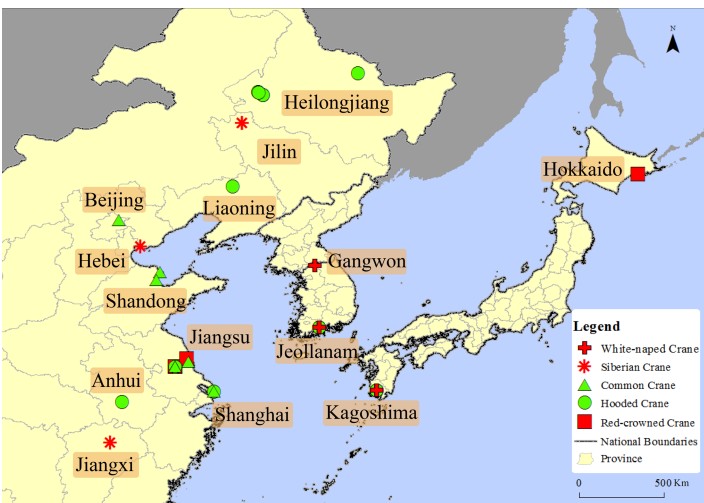

**Figure 1 The mix distribution map of sandhill cranes in Asian during the non-breeding period.** This figure depicts the mixed position of the sandhill crane with five other crane species in Asia. The cross represents the cranes mixed with the white-naped crane, the snowflake represents the cranes mixed with the Siberian crane, the triangle represents the cranes mixed with the common crane, and the square represents the cranes mixed with the red-crowned crane.

white-naped crane, Siberian crane, hooded crane, Eurasian crane and red-crowned crane (see Fig. 1).

Regarding whether it was a subadult, there were three clear records showing that a subadult was mixed alone with other crane species. These records appeared in 1969, 1984 and 2014, respectively, in Hokkaido, Jiangxi, and Hebei in Japan and China; one record showing that a family group was mixed with other crane species; 18 records showed an adult mixed with other crane species; and seven pieces of data were missing this kind of information.

As for the type of group, the records of the sandhill cranes mostly consisted of 1~3 individuals that were not a family group. In recent years, there were two records showing that there were already sandhill crane family groups in Asia during the non-breeding period. One family group was recorded in Yancheng National Wetland National Nature Reserve for Rare Birds, Jiangsu, China on December 4, 2013, with two juveniles in the reserve. The second was recorded in Kagoshima, Japan, on November 7, 2014, also with two juveniles recorded.
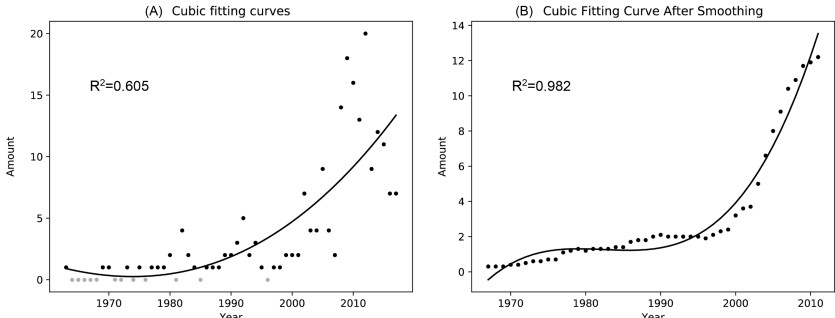

**Figure 2** **Cubic polynomial fitting curve of the number of sandhill cranes in Asia during the non-breeding period from 1963 to 2017.** (A) is a scatter plot of the number of the sandhill cranes in Asia in the past half century and its cubic fitting curve; (B) is a scatter plot of the number of the sandhill cranes in Asia in the past half century and its cubic fitting curve after performing a simple mean value filter.

## Analysis of individual amount changes

A total of 86 pieces of data were retrieved from 1963 to 2017 after de-duplicated. Finally, 98 pieces of data were obtained for the statistical analysis of the individual amount changes in successive years after supplementing the years without sandhill crane observation records as zero. Since the sample determination coefficient of the three, four and five polynomial fitting results after the filter processing showed no significant differences (0.007), to avoid overfitting, no higher-order polynomial fitting was performed in this study, and the three polynomial fitting results were selected. The results of the cubic polynomial fitting before and after the filtering process are shown in Fig. 2, and the polynomial fitting results for other powers are shown in Appendix, Fig. A2.

The results of the cubic polynomial fitting curve show that the amount of sandhill cranes during the non-breeding period in Asia increased significantly from 1963 to 2017. Before 1997, their quantity increased slowly with 0.06 individual per year. Since 1997, their amount grown rapidly with 2.53 individuals per year.

## Distribution change analysis

From 1963 to 2017, the non-breeding distribution area of the sandhill cranes in Asia reached its westernmost extent at the Yellow River Wetland National Nature Reserve (113°E) in Henan Province, China on November 11, 1999; the easternmost extent of the distribution was reached at Hokkaido (144°E), Japan on January 18, 1970; the southernmost extent was reached at Poyang Lake Nature Reserve (29°N) in Jiangxi Province, China on February 14, 1985; the northernmost distribution was reached at Xinqing wetland (48°N) in Heilongjiang Province, China, on September 18, 2009. The distribution spans 31 degrees in longitude and 19 degrees in latitude.

The distribution area of the cranes has increased over time. Thirty records showed that before the year 2000, sandhill cranes were distributed in 6 areas in Shuyang, Jiangsu; Yancheng, Jiangsu; Poyang Lake, Jiangxi; Yellow River Delta Nature Reserve, Shandong; and Yellow River Wetland Nature Reserve, Henan in China and Kagoshima and Hokkaido in Japan; Korea had no record during this period. An additional five areas were recorded in

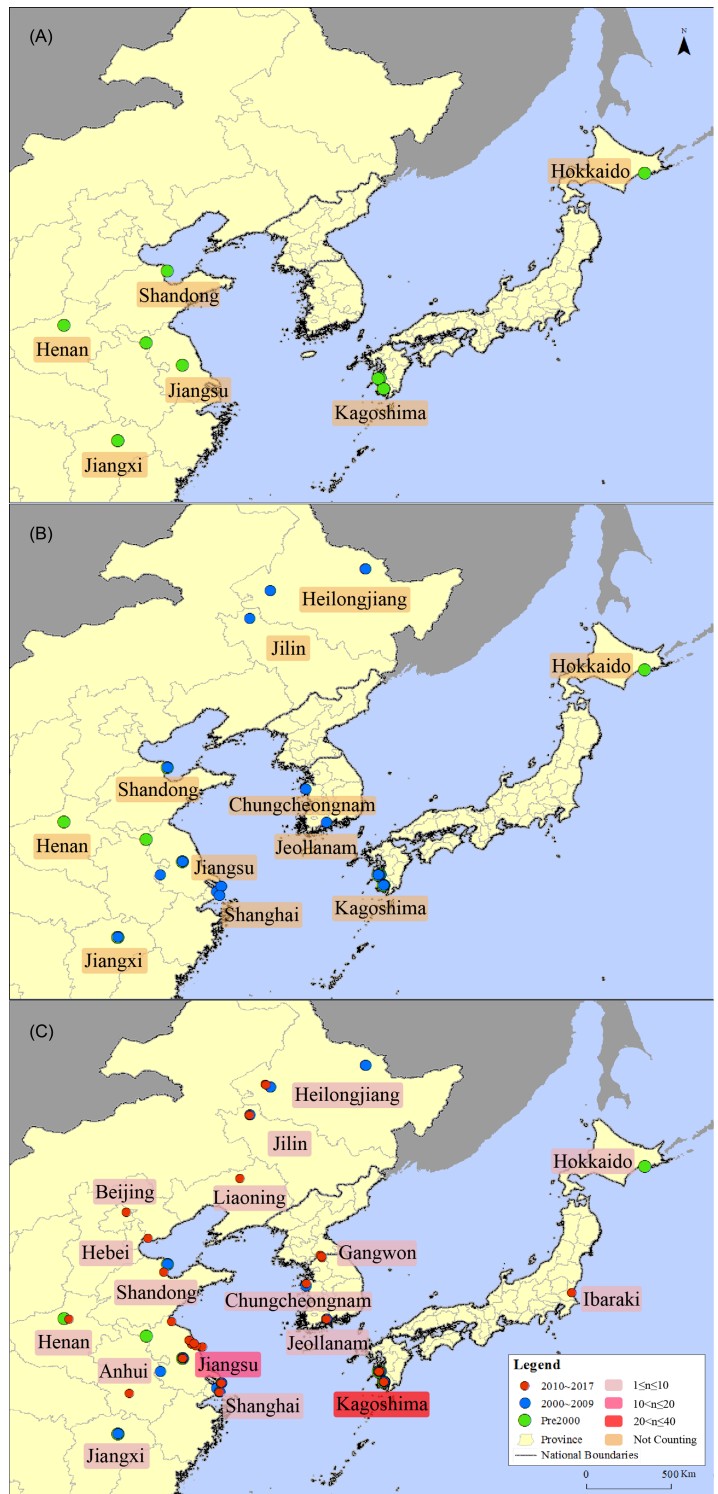

**Figure 3** **Distribution map of the sandhill cranes in Asia during the non-breeding period.** (A) is a schematic diagram showing the distribution of sandhill (continued on next page...)

**Figure 3 (…continued)**
cranes in East Asia during the non-breeding period before 2000; (B) is a schematic diagram showing the distribution of sandhill cranes in East Asia during the non-breeding period before 2009; (C) is a schematic diagram showing the distribution of sandhill cranes in East Asia during the non-breeding period before 2017. The red circle indicates the distribution point between 2010 and 2017; the blue circle indicates the distribution point between 2000 and 2009; and the green circle indicates the distribution point before 2000. The light pink box indicates that the number of distributed records in this area is from 1 to 10; the pink box indicates that the number of distributed records in this area is from 10 to 20; and the red box indicates that the number of distributed records in this area is from 20 to 40.

the following period from 2000 to 2009. There were 52 records showed that the distribution area of the cranes in China expanded to Zhalong National Nature Reserve, Heilongjiang; Momoge National Nature Reserve, Jilin; Xinqing Wetland, Heilongjiang; and Nanjing and Shanghai. Meanwhile, sandhill cranes were recorded in Suncheon Bay Wetland, Jeollanam and Seosan, Chungcheongnam in Korea. An additional six areas were recorded in the following period from 2010 to 2017. There were 87 records showed that their distribution area in China has expanded to Guanzidong wetland, Liaoning; Miyun Reservoir, Beijing; Lianyungang, Jiangsu; Tangshan, Hebei; and Caizihu, Anhui. Their distribution points in Japan increased to Ibaraki, and in Korea, their distribution increased to Cheolwon, Gangwon, see Fig. 3.

In China, sandhill cranes were distributed in 11 provinces and municipalities, including Heilongjiang, Jilin, Liaoning, Hebei, Beijing, Shandong, Henan, Jiangsu, Anhui, Shanghai and Jiangxi. In Japan, sandhill cranes were distributed in three provinces and cities, including Kagoshima in the south, Ibaraki in the east, and Kushiro, Hokkaido in the north. In Korea, sandhill cranes were distributed in Suncheon Bay Wetland, Jeonnam in the south and Seosan, Chungcheong and Ibaragi, Gangwon in the north.

## DISCUSSION

Sandhill cranes are the most numerous among all the species of cranes in the world and are still on the rise (*Wetlands International, 2018*). The sandhill cranes that breed in northeastern Russia have increased. From 1995 to 2009, their amount in a model site in tundra near Indigirka River in summer grew 13.3 times. The expansion trend is both in their amount and their distribution areas (Personal communication with Bysykatova Inga Prokopievna) (*Degtyarev, 2008*; *Prokopievna, 2012*).

The results of our study show that the distribution area of the sandhill cranes in East Asia during the non-breeding period was increasing, and the amount of individuals was also increasing each year (see Figs. 2 and 3). Therefore, their distribution and amount in Asia during the non-breeding stages may be undergoing a process similar to that in Russia. Before 1963, there were no records of sandhill cranes wintering in Asia (*Crane Park Izumi, 2017*). The reports before 2013 regarding the wintering records of the sandhill cranes were all sporadic individuals who mixed with other crane groups, indicating that these individuals may come to Asia together with other crane species by some accidental factors that occurred in the breeding grounds or the stopover sites. Therefore, it is reasonable to define sandhill cranes as ''vagrant'' (*Masatomi, 1972*; *Wang et al., 2007*; *Zheng, 2011*).

In the past decade or so, the frequency and location of records have increased (see Figs. 2 and 3). Sandhill cranes wintering in Asia represent almost a normal phenomenon that continues to expand. In the winters of 2013 and 2014, a family group was recorded in Yancheng, Jiangsu, China and Izumi, Kagoshima, Japan respectively. Both of these records were found in the regions with the largest amount of data, and also have relatively stable observation records of sandhill cranes. These individuals may come from northeastern Russia. They chose a relatively short path to winter in Asia. This is also in line with the time-minimization and energy-minimization strategies of migratory birds (*Zhao et al., 2017*). Although coming to Asia for the winter is more in line with time-minimization and energy-minimization strategies. However, due to the habit of migration and the role of old birds in the migration of birds, there were no large amount of sandhill cranes recorded during the winter in Asia (*Mueller et al., 2013*). Although the amount is now few, the definition of "vagrant" is insufficient to describe their current wintering distribution status based on the long-term stable records (see Figs. 1 and 2). Based on this study, we defined the status of the wintering of sandhill cranes in China as "rare winter migratory birds" (*Cheng et al., 2014*) and suggested that the sandhill cranes in East Asia (China, Japan, and South Korea) also be defined as "rare winter migratory birds". The records of the family group of sandhill cranes in Asia in winter indicate that there are adult birds who lead the juveniles to migrate to Asia now. We assume that there may be more family groups that show up in the future and then expand their wintering groups in Asia to finally form a stable wintering population. Sandhill cranes will become a real "winter bird" in Asia. It's worth mentioning that the inevasible limitation of data used in this research, such as (1) potential misidentification between sandhill crane and other crane species; (2) increasing recorded sandhill crane individual numbers possibly influence by rising number of bird watchers, and (3) those observation records without exact individual amount were defined as only "1" individual in the data analysis. In the future, it could be further supplemented by the development of professional abilities (species identification and data collection) of bird watchers. It is also important to conduct long-term continuous monitoring in key wintering sites of sandhill crane in East Asia.

There were three clear records of subadult individuals wintering alone with other crane species in Asia, the earliest in 1969. There may have been additional occurrences of subadult wintering in East Asia that were not detected because of limitations of field identification abilities. That most of the records of subadult were sporadic (1969, 1984, and 2014) may reveal the role of subadult in the winter range expansion of sandhill cranes. The subadult distribution records show that subadults were recorded as early as 1969, and most of their occurrences were rarely recorded again in the future, which may also reveal the role of subadult in the expansion of sandhill cranes. The records of the subadult individuals are valuable in judging the spread mechanism of cranes. According to a study on the spread of the greater flamingo (*Phoenicopterus ruber*) in China, subadults were observed in high proportions in their expansion areas (79.2%) (*Zhu et al., 2017*). Therefore, subadults may be the initiators and implementers of the spread of the greater flamingo. From the three subadult records in this paper, it is difficult to determine whether the increase in occurrence of sandhill cranes in Asia is similar to the proliferation of the greater flamingo,

and it remains to be further studied. In addition, if the mixture of sandhill and other crane species individuals is selective or opportunistic might also relevant to its expansion mechanism and deserves more studies in the future.

Regarding the distance of the migration path, sandhill cranes that breed in Russia may have more advantages wintering in Asia than in their traditional wintering grounds in North America. The shorter distance in East Asia may reduce energy consumption during migration (*Zhao et al., 2017*). In addition, East Asia is an important wintering ground for many kinds of cranes. Sandhill cranes have similar habitat requirements to other crane species, such as the Eurasian crane and the red-crowned crane, in non-breeding seasons. Therefore, East Asia can provide them with adequate food and suitable habitats (*Swengel & Carpenter, 2000*; *Cheng et al., 2014*). Meanwhile, sandhill cranes can adapt to locally available herbivorous food resources, especially grains (*Johnsgard, 1983*). Good adaptability is also the basis for the species' further spread in Asia. With the increase of sandhill cranes in Asia during winter, there may be more sandhill cranes following the experienced old individuals migrating to Asia (*Mueller et al., 2013*). The wintering distribution of sandhill cranes in Asia may develop into a forward and directional expansion instead of following the other crane species.

The increase in sandhill cranes in Asia to become a winter bird may cause the following potential problems. First, in terms of disease transmission, sandhill cranes have the ability to carry and spread infectious diseases between North America and Asia. Previous studies have shown that sandhill cranes and other waterbirds can transmit highly pathogenic avian influenza viruses between Asia and North America (*Winker et al., 2007*; *Winker & Gibson, 2010*). Meanwhile, the results of the study that focused on the two large populations of sandhill cranes in North America showed that the population (MCP) that will migrate to Asia has a higher chance of being exposed to pathogens and parasites. Coccidian (*Sphaerozoum fuscum*) can be disseminated to the endangered whooping crane by sandhill cranes (*Hensel et al., 2017*). Highly pathogenic avian influenza infection and other diseases derived from coccidia and coccidiosis are all potential future impacts of sandhill cranes on other crane species in Asia. Second, since there have been records of field hybrids of sandhill cranes with whooping cranes (*G. Americana*) and Eurasian cranes, sandhill cranes may hybridize with other crane species (record from ebird, http://www.ebird.org). Genetic studies have shown that compared to white-naped Cranes, whooping cranes, hooded cranes, Eurasian cranes, Siberian cranes and demoiselle cranes, sandhill cranes have the closest genetic distance to red-crowned cranes (*Guo, Zhang & Song, 2012*). Therefore, it remains to be seen whether there will be hybrids of sandhill cranes and what it means for the endangered (EN) red-crowned cranes in the future (*IUCN, 2017*).

At the point of interspecific competition, studies have shown that the massive breeding of sandhill cranes in northeastern Russia has already exerted competitive pressure on Siberian cranes (*Germogenov et al., 2015*). During the nonbreeding season, most species of cranes have fairly similar foraging habits (grains, seeds, tubers), and sandhill cranes are known to be one of the most generalist species (*Reinecke & Krapu, 1986*; *Jesús, Juan & Deseada, 2002*; *Zhao, Ma & Chen, 2002*). Therefore, they are likely to overlap in habitat and foraging with other crane species in Asia. That might be an important influence that

sandhill crane will bring. The further influence of these pressures on endangered cranes remains to be further studied. Because the sandhill cranes were mixed with five other species of cranes in Asia during the non-breeding period (Fig. 1), once they have released their huge potential expansion ability, they will have a serious impact on those species that have long been distributed in Asia, especially for the red-crowned cranes, Siberian cranes, hooded cranes and white-naped cranes. These impacts are what we ought to focus on, and they should be prevented in advance.

## CONCLUSIONS

Since the first record of sandhill crane wintering in Asia in the 1960s, its wintering distribution area here has expanded, and its number of individuals has also increased over time. This phenomenon is closely related to the steady growth trend of its source population in North America and the growing and expansion trend of them during summer time in Asia. In recent years, family groups of sandhill cranes has been recorded wintering in Asia, and not just a single individual anymore. According to the current situation, the wintering individuals in Asia have a potential tendency of increasing. Once it happens, it will have a series of effects on other crane species that are wintering in the same region in Asia.

Up to now, there has been no designed field investigation for sandhill cranes. Our research has clarified the situation of sandhill cranes wintering in Asia since the first records. But research on its wintering status is still difficult to conduct. In the future, we can predict the possible wintering area by using the species distribution model to narrow the scope of research and conduct further study on it. Related researches are of great significance to the study of the expansion mechanism in a new environment of the crane which has the largest number of individuals around the world. It can also estimate the possible impact of the expansion of sandhill cranes and to develop appropriate management approaches on it.

## ACKNOWLEDGEMENTS

We thank the GBIF team for making their database available freely and all the people who contributed to the open access data. We appreciate Bysykatova Inga Prokopievna for providing the manuscripts about sandhill cranes in Russia. We greatly thank Huajin Sun in Yancheng, China, for supplying us with his photographic records in Yancheng. We thank Xiaolong Liu, Jun Wang and Chuyu Cheng for comments on previous versions of this manuscript. We are grateful to Falk Huettmann and Qi Cong for their suggestion on our analysis methods.

### Funding

This work was supported by the National Natural Science Foundation of China awarded to Yumin Guo (NO. 31570532). The funders had no role in study design, data collection and analysis, decision to publish, or preparation of the manuscript.

## Grant Disclosures

The following grant information was disclosed by the authors:
National Natural Science Foundation of China: 31570532.

## Competing Interests

The authors declare there are no competing interests.

## Author Contributions

- Linqiang Gao conceived and designed the experiments, performed the experiments, analyzed the data, contributed reagents/materials/analysis tools, prepared figures and/or tables, authored or reviewed drafts of the paper, approved the final draft, submission & Revision.
- Chunrong Mi conceived and designed the experiments, analyzed the data, authored or reviewed drafts of the paper, approved the final draft.
- Yumin Guo conceived and designed the experiments, performed the experiments, contributed reagents/materials/analysis tools, authored or reviewed drafts of the paper, approved the final draft.

## Data Availability

Data is available in the Appendices, the Supplemental File and Zenodo: Guo Yumin. (2019). The distribution data of sandhill crane in Asia during non-breeding period from our fieldwork [Data set]. Peerj. Zenodo. http://doi.org/10.5281/zenodo.3361856.

## Supplemental Information

Supplemental information for this article can be found online at http://dx.doi.org/10.7717/peerj.7545#supplemental-information.

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
