# Peer review of "Expansion of sandhill cranes (Grus canadensis) in east Asia during the non-breeding period"

_PeerJ, doi:10.7717/peerj.7545_

## Round 0.1 · original submission · Minor Revisions

Based on two very detailed reviews, I am pleased to request minor revisions on your paper. Please note, each of the two reviews are very helpful and should greatly improve the value of this paper. Of special note, please clarify terminology throughout the manuscript, in particular your definition of filtering and terms about bird age classes. I look forward to your revision. Very nice work.

Reviewer 1 ·

Basic reporting

.

Experimental design

.

Validity of the findings

.

Additional comments

[Also see attachement]

Lines – statements

14. “Historically, the distribution of Sandhill Cranes included much of North America and extending in summer into northeast Russia.”
18. “spread south into East Asia”.
19. “and to assess…”
31. “with five other species of cranes”.
32. “that the range and numbers of Sandhill Cranes have expanded”.
33. “were mixed with five other species of cranes”
35. “ advantages of these other species”
36. “other species of cranes”
37. “current perhaps the most abundant of th 15 species of cranes”.

Figure 1. Sandhill Cranes are with both White-naped and Red-crowned on the Korean DMZ, with both Siberian and White-naped at Poyang Lake. Perhaps there are other species in contact with them at the other locations indicated. There is much overlap with the distribution of many crane species during migration and in winter in East Asia, with the exception of Red-crowned in Hokkaido. If the data show association with specific species, that should be discussed if possible as selective or opportunistic.

Figure 2. It should be mentioned that the number of bird watchers has also increased since the 1960s (China was in the throes of the Cultural Revolution then). The bird watching groups in the three nations might have information on their numbers to graph with the sightings of Sandhills.

Annotated reviews are not available for download in order to protect the identity of reviewers who chose to remain anonymous.

·

Basic reporting

The structure, figures, and table are good, and the raw data supports the results and discussion. The abstract provided a succinct summary of the results. References are appropriate and complete, and provide good background and context.
Professional, technical English is used, but there are a number of areas where better word choice or wording is needed for clarity. Some suggested wording changes and problematic areas:
Line 25 and in abstract: Replace ‘Python” (the software) with regression analysis; specifics of the software seems unnecessary since this is a common statistical method.
Line 27: insert “during the non-breeding season in Asia” following sandhill crane to clarify the distribution details and season.
Line 34-35: During the nonbreeding season, most species of cranes have fairly similar foraging habits (grains, seeds, tubers), and sandhill cranes are known to be one of the more generalist species. Suggest changing / simplifying this statement to ~ “indicating their adaptability to a range of habitat types and food resources.”
Line 36: “Future development trends” could refer to human developments. Recommend changing wording of this sentence to “The implications of these trends in sandhill cranes in East Asia for this and other crane species warrants further research”
Line 51: Phymer et al. should be Rhymer et al.
Lines 47-61: Data in this paper deals sandhill cranes associated with the Midcontinental Population, and specifically the less subspecies; it not necessary to name the other North American populations. I think this pargraph could be rewritten to indicate that it is the lesser sandhill crane subspecies that migrates to Asia from central North America, that it is part of the largest population of sandhill cranes, and that population continues to increase in number and its breeding range in Asia.
Line Lines 63-64: Should clearly indicate in this sentence that these are birds that fly to Russia for the breeding season.
Line 76: Clarify what region here – time of arrival to different breeding grounds generally, or here specifically about North American breeding grounds?
Line 89: “Sandhill cranes wintering in East Asia has long been reported.” Do you mean back to 1963, or were there earlier anecdotal reports? If back to 1963, these first 2 sentences could be merged to “have been reported since 1963, with first reports in Izumi….”
Lines 168 and 174: suggest it is more appropriate to use “created” rather than “obtained” as the data were used to create or generate maps and table.
Lines 208-212: suggested revision for better flow: “In recent years, there were 2 records showing that there were already sandhill crane family groups in Asia during the non-breeding period. One family group was recorded in Yancheng National Wetland National Nature Reserve for Rare Birds, Jiangsu, China on December 4, 2013, with 2 juveniles in the reserve. The second was recorded in Kagoshima, Japan, on November 7, 2014, also with 2 juveniles recorded.”
Line 262: “Previous reports” – previous to when? 1963 or a report before this paper?
Lines 288-291: Based on my assumption that “subadult” means a bird <1 year old (‘young’ in the raw data), I suggest the following revisions for this paragraph: “There were four clear records of subadults individuals wintering alone with other crane species in Asia, the earliest in 1969. There may have been additional occurrences of subadults wintering in East Asia that were not detected because of limitations of field identification abilities. That most of the records of subadults were sporadic (1969, 1984, 2011, and 2014) may also reveal the role of subadults in the winter range expansion of sandhill cranes.”
Line 301: Insert “that breed in Russia” after sandhill cranes to differentiate from North American breeding cranes.
Lines 304+: As indicated for the Introduction, worth noting that sandhill cranes are generalists in food habits, and therefore are likely to overlap in habitat and foraging with other crane species.
Lines 315: Recommend dropping “dangers”; simply” potential problems” is sufficient here. When cranes are beginning the courtship and pairing relative to their associations with other crane species or their own will be important in evaluating to what extent this is a real risk to Asian crane species.
Lines 345-347: Duplicated sentence to the following sentence; delete one of the 2
Lines 347-348: “In two recent years, a family of sandhill cranes…” Recommend dropping “and not just as a single individual mixed with other cranes” as this phrase is confusing.
Throughout: Often the authors refer to ‘other kinds of cranes.’ Recommend consistently referring instead as ‘other crane species’, to differentiate from types of groups (e.g, singles, family, grouped adults). Also, please change “common crane” to “Eurasian crane” throughout – this is the accepted common name. Siberian should always be capitalized in the name of that species.

Experimental design

This is an observational study, but one which takes full advantage of a range of reporting sources as well as recent field work. The research question is clearly stated and then addressed in the results and discussion. The statistics applied were appropriate to elucidate the temporal pattern of numerical increase. Additional clarification is needed to demonstrate rigor in terminology, how some of the data sources were assessed, the data filtering for analyses, and to be able to replicate both field and searches for existing records.
Lines 139-140: More explanation is needed to demonstrate how your “understanding of cranes” was used to filter information from news and bird watching sites when photographs were not available. For example, how could you determine if there was a mis-identification of a species when reported as a sandhill crane but no photograph was available?
Lines 152-169 – Data Filters: Several points need to be clarified here --
1) Subadults (first used on line 161): Please define subadults and how this age was identified in observations. Is a “subadult” defined a bird between 1~3 years? Or a bird <1 year old? How was the age determined -- based on plumage, social grouping, behavior? In the raw data there is no “subadult” column, but there is a “Young” column, which I am assuming for this review is equivalent to “subadult” in the text. But families are described as adults and 2 “juveniles” – what is the difference if subadults are <1 year old? Clarifying terms for subadults and juveniles here and throughout the ms is very important to understand the differences and understanding how sandhill cranes are ending up wintering in East Asia. Review the paper throughout to be clear and consistent. In my review of the results and conclusions, I came to assume subadult = young = a sandhill crane <1 year old.
2) Year: How year was assigned was not immediately clear to me until I reviewed the data. I recommend changing the wording to indicate that if the month of observation was in January-May, then the record was assigned the previous year, to indicate the bird(s) was considered to have been present in East Asia that entire winter (fall-spring).
3) Mixed With – the data indicate either other crane species that the sandhill crane(s) were observed with, or “NA”. NA accounted for 139 of 168 observations. NA could be interpreted as simply no data, or (considering the general interest in cranes for most of these sources) that the sandhill cranes were not with other cranes. Please clarify so readers can appropriately interprete these records.
Line 199: It is not clear what is meant by “no separate activity records”. Since this in the section about misxed, subadult, adult and family records, you mean sandhill cranes were only observed with other cranes, and never reported away from other birds (on their own)? Or were never observed as a solitary sandhill crane – e.g., “…with other cranes” or other crane species? Review other sentences in the results and discussion to ensure clarity of other species versus simply other cranes.
Lines 214-215: In reviewing the data table, I am confused on how these numbers were derived. There are 144 records with the number of cranes recorded (remainder are NA). In this sentence, the second number (98) – after the filter was applied – is higher than the original number of data (86). How is that possible? Please explain the filter used to achieve the 86 and 98 records for regression analysis, and what each number represents.
Supplementary materials: In Table 1-- 1) Recommend dropping the parentheses in the right column around the citations, as they are unnecessary here. 2) “Bird Talker 2018” appears to be “Talker, B.” in the reference list; recommend changing in the references as Bird Talker”. 3) NaverBlog 2006 is not listed in the references.

Validity of the findings

The data is robust and support the results and main conclusions The paper provides valuable insights, and foundational information, about an expanding species, which is shifting from ‘winter vagrant’ toward a more common wintering species in East Asia. The risk of greater disease transmission is probably the most important challenge to Asian cranes during the nonbreeding season; suggest elevating that to be the main part of that portion of the discussion. Implications of hybridization is likely less serious to the future overall health and success of native Asian cranes, and will likely be uncommon. Some other suggestions for improvement:
Lines 224-224: Please express the increase as percentages rather than simply “significantly” to more clearly demonstrate the magnitude of change.
Lines 234-244: Would be helpful to indicate, in each time period, the number of areas where sandhill cranes were observed (e.g., 7 areas before 2000, an additional xx records in the following periods).
Line 255: as noted above, would be helpful to indicate percent increase (ie magnitude of change), and give numbers – increasing from individual cranes in a single area before 1980 to as many as 8 birds ….
Lines 349-305: Confusing sentence that has little value beyond the obvious; recommend deleting.

Additional comments

The paper lays out the historic and ecological background for the research question in the introduction, and the appropriate literature are referenced. The raw data provided were clear and very useful in reviewing and understanding the results. Results support the main conclusions. The pattern reported of progressive increase in numbers and social groups is valuable to understanding how these birds may be shifting their migratory habits, with longer term implications for the development of a new migratory pathway and tradition. This paper will provide useful foundational information about the status and trends of sandhill cranes wintering in East Asia.
Below are more specific questions and comments on parts of the text. Generally, key points to address are:
1. Define or clarify terms used there, in particular subadult versus juvenile.
2. Improve the explanation of how the raw data were filtered to obtain the final set used for regression.
3. Ensure consistency in the use of terms throughout the paper (e.g., subadult, young, amount; other cranes versus other crane species) and be sure it matches the raw data table (young = subadult? amount = number).

---

## Round 0.2 · accepted · Accept

Based on the revised submission and a secondary review from one of the reviewers, I am happy to accept this publication. Congratulations.

·

Basic reporting

The authors have addressed all my earlier concerns and questions with appropriate changes. While some of the English phrasing is a bit awkward and might benefit from some minor additional editing, the meaning is clear.

Experimental design

No comment

Validity of the findings

No comment